# Anionic Methacrylate Copolymer Microparticles for the Delivery of Myo-Inositol Produced by Spray-Drying: In Vitro and In Vivo Bioavailability

**DOI:** 10.3390/ijms25073852

**Published:** 2024-03-29

**Authors:** Roberto Caruana, Maria Grazia Zizzo, Gaetano Felice Caldara, Francesco Montalbano, Silvia Fasciano, Dora Arena, Marida Salamone, Gaetano Di Fazio, Alessandro Bottino, Mariano Licciardi

**Affiliations:** 1Technology Scientific S.r.l., Via del Quarnaro 14, 90144 Palermo, PA, Italy; roberto.caruana@gmail.com (R.C.); f.montalbano@tecscien.com (F.M.); 2Dipartimento di Scienze e Tecnologie Biologiche Chimiche e Farmaceutiche (STEBICEF), Università degli Studi di Palermo, 90123 Palermo, PA, Italy; mariagrazia.zizzo@unipa.it; 3Dipartimento di Promozione della Salute, Materno-Infantile, di Medicina Interna e Specialistica di Eccellenza “G. D’Alessandro”, Università degli Studi di Palermo, 90123 Palermo, PA, Italy; gaetanofelice.caldara@unipa.it; 4IDI Integratori Dietetici Italiani S.r.l., Via G. Mameli 12, 95020 Aci Bonaccorsi, CT, Italy; silvia.fasciano@idipharma.com (S.F.); dora.arena@idipharma.com (D.A.); marida.salamone@idipharma.com (M.S.); gaetano.difazio@idipharma.com (G.D.F.); alessandro.bottino@idipharma.com (A.B.)

**Keywords:** myo-inositol, spray-drying, controlled release, microparticles, polycystic ovary syndrome (PCOS), anionic methacrylate copolymer

## Abstract

In this study, a new micro delivery system based on an anionic methacrylate copolymer, able to improve the biological response of myo-inositol by daily oral administration, was manufactured by spray-drying. It has an ideal dose form for oral administration, with an experimental drug loading (DL)% of 14% and a regulated particle size of less than 15 µm. The new formulation features an improvement on traditional formulations used as a chronic therapy for the treatment of polycystic ovary syndrome. The microparticles’ release profile was studied and ex vivo porcine intestinal mucosa permeation experiments were performed to predict potential improvements in oral absorption. Batch n. 3, with the higher Eudragit/MI weight ratio (ratio = 6), showed the best-modified release profiles of the active ingredient, ensuring the lowest myo-inositol loss in an acidic environment. The in vivo evaluation of the myo-inositol micro delivery system was carried out in a rat animal model to demonstrate that the bioavailability of myo-inositol was increased when compared to the administration of the same dosage of the pure active ingredient. The AUC and Cmax of the loaded active molecule in the micro delivery system was improved by a minimum of 1.5 times when compared with the pure substance, administered with same dosage and route. Finally, the increase of myo-inositol levels in the ovary follicles was assessed to confirm that a daily administration of the new formulation improves myo-inositol concentration at the site of action, resulting in an improvement of about 1.25 times for the single administration and 1.66 times after 7 days of repeated administration when compared to pure MI.

## 1. Introduction

Polycystic ovary syndrome (PCOS) is an endocrine disorder, affecting women throughout their lifetime. The available literature regarding the prevalence in Europe reports a broad range between 2.3% and 28% [1,2,3,4,5], triggered by multiple factors including from genetic inheritance [6], lifestyle, and environmental stimuli [7,8]. The syndrome generally results in a clinical frame characterized by hyperandrogenism, ovulatory dysfunction, polycystic ovaries [9], and insulin resistance. This latter condition has gathered the attention of several research groups due to its involvement in several metabolic abnormalities comprising visceral obesity and hypertension [10]. It occurs in PCOS-affected women more than in healthy controls with the same age and Body Mass Index (BMI) [11] and it results in compensatory insulin resistance [12].

Myo-inositol (MI) is a stereoisomer of a C6 sugar alcohol that belongs to the myo-inositol family [13]. It is the precursor of myo-inositol triphosphate, an intracellular second messenger regulating several hormones such as thyroid-stimulating hormone, follicle-stimulating hormone (FSH), and insulin [14]. The exact mechanism of action of MI as an insulin mimetic is still unclear [15]. One of the most interesting models predicates [16] that insulin binding to its receptor activates epimerase, converting myo-inositol into D-chiro-inositol (DCI). A second pathway involves a G protein bound to the insulin receptor, binding a phospholipase that catalyzes the hydrolysis of a glycosylphosphatidylinositol (GPI) [17]. The insulin-induced hydrolysis of the GPI releases a myo-inositol phosphoglycan containing DCI (DCIIPG), which acts as a probable second messenger of insulin (INS-2), mediating insulin’s effects on glucose’s oxidative and nonoxidative clearance. Moreover, it is reported that part of the MI supplementation effect comes from its partial intracellular epimerization to DCI [18]. The use of inositol(s) isomers in the management of polycystic ovary syndrome has a relevant clinical impact [19,20]. 

However, traditional MI supplement formulations are unable to ensure reproducibility in terms of oral absorption and bioavailability, giving sub-therapeutic concentrations of the active molecule. This can prevent biological responses from being triggered, leading to an inconsistent therapeutic result. To trigger similar therapeutic responses in heterogeneous cohorts, the dosage form must ensure that the concentration of the active molecule at the glycocalyx in the enteric epithelium is similar in the largest portion of the tested population [21,22,23]. To achieve this, the dosage form should be designed to minimize the loss of the initial dosage and to maximize the solubility of poorly soluble molecules or the bioavailability of soluble molecules. Traditional supplement dosage forms usually do not pursue this goal, nor do they prevent the loss of active molecules, for instance, by degrading agents (H_3_O^+^, digesting enzyme); off-target release; or the burst effect. 

The extent of these processes influences the efficacy of marketed supplements with no delivery strategy, for which a different effect in each individual may be expected [22]. On the other hand, gastrointestinal side effects can be observed when a high MI dose is administered to overcome the low bioavailability of the active molecule [24]. For all of these reasons, we developed a novel micro delivery system using spray-drying technology, aiming not only to enhance the bioavailability of MI but also to mitigate the variability in biological response typically observed with its oral administration. In the past, an attempt to improve MI bioavailability was made by proposing the use of a soft gel capsule as a formulation strategy, demonstrating that a capsule containing 0.6 g of MI was equivalent to the pharmacokinetic parameters of 2 g of MI in powder form [22]. In this study, a new MI delivery system was manufactured by spray-drying, featuring not only increased bioavailability but also delayed (gastro-resistant) and prolonged (intestinal) release when compared with the pure active ingredient. The key innovation of our work lies in the use of Eudraguard Biotic, an anionic copolymer of methyl acrylate, methyl methacrylate, and methacrylic acid, in a 7:3:1 ratio, respectively, as a co-formulant in the spray-dried formulation. This commercially available gastro-resistant excipient has been approved by the European Commission as a safe food additive and is commonly used as a coating material for delayed release in oral dosage forms [25,26]. By leveraging the unique properties of Eudraguard Biotic, we achieved not only increased bioavailability but also delayed and prolonged release characteristics compared to the pure active ingredient. In summary, our study represents a contribution to the field by introducing a novel micro delivery system for myo-inositol, which not only improves its bioavailability but also addresses key limitations associated with its oral administration. Through the strategic use of Eudraguard Biotic as a co-formulant and the spray-drying technique as a microparticle production method, we achieved a formulation with enhanced efficacy, safety, and potential for clinical translation.

## 2. Results

In this study, different batches of micro delivery systems loaded with MI were prepared to optimize the spray-drying process, yields, DL, and delayed release profile (Table 1) of the tested microparticle formulation. The optimization of process parameters was centered on varying the Eudraguard/MI weight ratio and, if necessary, the spray-dryer operating parameters, such as temperature and air- and feed-flow values. Subsequently, various production attempts were carried out, until the product with the best qualities in terms of yield, DL, and release profile was obtained. The process yields ranged from 40 to 50%. Such low yields are common when using lab-scale spray-dryers, widely reported in the literature [27] when methacrylate copolymers are involved [28]. The drug loading of MI ranged between 14 and 47% by varying the Eudragard/MI weight ratio from six to one, respectively. The incorporation efficiency values (IE%) were very high in all three batches; in particular, batch 3 a value of 100% incorporation efficiency of myo-inositol. The set of all of these parameters (yield%, drug loading%, incorporation efficiency%, Eudragard/MI weight ratio) and the release studies at pH 1 led us to select batch no. 3 (in Table 1), the one with the slowest release profile in an acidic environment, for further biological evaluations.

In all cases, SEM analysis showed a quite homogeneous spherical shape of all produced microparticle batches, with an average diameter size of 13.5 ± 5.7 μm (Figure 1).

For this study, the release of the active molecules were assessed in vitro by a pH jump test [29]. Despite containing the Eudraguard Biotic, a gastro-resistant synthetic excipient, the MI-containing microparticles released more than 50% of the MI at gastric pH in two hours. In particular, batch n. 3, with the higher Eudragit/MI weight ratio (ratio = 6), showed the slowest release profile (Figure 2), and for this reason, was the only one used for further biological evaluations.

### 2.1. Franz Cell Porcine Colon Mucosa Permeation Model

The batch n. 3 MI-loaded micro delivery systems (MPs, 50 mg, DL = 14%) were tested in a permeation study to assess a potential increase in drug permeation when compared with the pure MI, using the permeation of a membrane made of porcine colon mucosa in a Franz cell permeation model as the metric. For this purpose, two Franz cells were used simultaneously, set up as shown in Figure 3, and the experiment was repeated three times [29,30].

This study showed that the micro delivery system (MPs) prolonged the permeation time and increased the cumulative amount of permeated MI (area under the curves, AUC) by about three-fold (Figure 4) if AUCs are compared (AUC MPs = 4.86; AUC Inositol = 1.65).

### 2.2. In Vivo Bioavailability and Ovarian Accumulation

As expected, the in vivo bioavailability of MI also increased when orally administered as microparticles. Figure 5 shows the values of the plasma concentration of MI following its administration, both in its pure form (group 2) and delivered by the new microrelease system, batch n. 3 (group 3). Each point of the curves (MI plasma concentration vs. time) in Figure 5 represents the average values of the plasma concentrations of MI; the respective AUC values and the relative ratio are shown in Table 2.

The improvement in bioavailability after a single MP administration enabled the prediction of a potential increase in MI concentration at the site of action, such as the ovarian follicles. The above prediction was demonstrated by evaluating the MI concentration in the rats’ follicles upon single and multiple administration of microparticles. Each MI concentration value was determined in biological tissues (ovaries) as the average of the MI tissue concentration values in the treated animals (animal groups 5 and 6). The bar graph in Figure 6 shows the concentration of MI in the ovaries after 24 h (single administration) and after 7 days (administration once a day for 7 days), compared to the basal concentration of MI in the tissue of interest. Table 3 shows the calculated concentration values.

## 3. Discussion

Spray-drying is a technique providing many benefits, such improving the dissolution, absorption, and therapeutic efficacy of drugs compared to traditional dosage forms, in the manufacturing of micronized active formulations and multicomponent solid dispersion [31,32,33]. The advantage of using this production technique in this case is being able to obtain a solid oral formulation with unconventional release characteristics in a single step. Gastro-resistant polymers are commonly used for the coating of pre-formed solid formulations, resulting in a further production step and higher production costs.

A controlled-release pattern is a key issue in the development of nutraceutical delivery systems. Myo-inositol is a molecule that is highly soluble in water, even in its crystalline form; it is a powerful osmotic agent and partially soluble in many polar organic solvents [34]. This characteristic makes it very difficult to slow down the release rate from an oral formulation or generate prolonged release formulations. We can therefore consider it a good achievement to have produced a formulation that allows us to convey almost half of the dose administered to the intestine.

When tested in vivo, the MI-loaded micro delivery system exhibited a higher bioavailability and a higher concentration in the follicles. From the calculation of the area under the curve (AUC), it can be seen that the new micro delivery system batch n. 3 provides a greater bioavailability of MI compared to the same active ingredient in pure form. The AUC and Cmax of the loaded active molecule were improved by a minimum of 1.5 times when compared with the pure substance, administered with the same dosage and route. The results demonstrate that MI, when conveyed by the new microrelease system, is absorbed to a greater extent than pure MI, mainly in the intestine, as partially evidenced by the visible concentration jump in the bioavailability curve (Figure 5) starting from about 2 h. Pure MI, on the other hand, is immediately absorbed in the stomach, although to a lesser extent than the microparticles, and consequently, it is eliminated from the plasma faster. The report of the respective AUC confirms what was predicted by the in vitro permeation studies, namely that the new release system increases both the overall bioavailability and the half-life of MI. This is due to the modulation of the release of MI, which is partially site-specific and prolonged, thus dictating a slowing down and extension of the absorption, which significantly increases the peaks of plasma concentrations and extends the half-life of MI.

The biodistribution results demonstrated that, upon chronic administration, MPs provide high MI levels in the follicle of the animal models used in the present study, resulting in an improvement of about 1.25 times for the single administration and 1.66 times after 7 days of repeated administration, when compared to pure MI (Table 3).

In conclusion, the new myo-inositol release system constitutes a clear technological advance in the dosage forms of MI, capable of increasing the bioavailability, and could enable the dose to be reduced or the dose intervals to be prolonged in chronic therapy with MI.

## 4. Materials and Methods

### 4.1. Materials

Myo-inositol (MI) was provided by IDI (Integratori Dietetici Italiani S.r.l., Aci Bonaccorsi, Italy); Eudraguard Biotic was purchased from Evonik, Germany; the commercial Eudraguard Biotic is available from the producer as 30% (*w*/*v*) aqueous dispersions and contains various amounts of surfactants as emulsifying agents, specifically sodium lauryl sulfate and polysorbate 80 [23]. HCl was purchased from VWR; a myo-inositol assay kit, K-INOSL, was purchased from Megazyme, Chicago, IL, USA.

### 4.2. Animals

Thirty Wistar female rats (weighing 250–350 g) were purchased from ENVIGO S.r.l. (San Pietro al Natisone, UD, Italy) and were employed throughout this study. The animals were housed in temperature-controlled rooms on a 12 h light cycle at 22–24 °C and 50–60% humidity. They were fed standard laboratory chow and tap water ad libitum. The animals were allowed to acclimatize to the housing conditions for 1 week before experimentation. Procedures involving the animals and their care were conducted in conformity with the Italian D.Lgs 26/2014 and the European directives (2010/63/EU). Animal care and handling were conducted following the provisions of the European Community Council Directive 210/63/UE, recognized and adopted by the Italian Government. The experiments were approved by the Ethical Committee for Animal Experimentation of the University of Palermo and by the Italian Ministry of Health (authorization n. 813/2019-PR). No other methods to perform the described experiments (3Rs) were found. 

### 4.3. Preparation of the Micro Delivery Systems

The microparticles of myo-inositol were prepared with a Mini Spray Dryer Buchi B290 (Buchi, Switzerland). The spray-drying process was operated according to the following parameters: inlet T: 110 °C; outlet T: 60 °C; 100% suction; feed pump: 15%; atomizer nozzle: 0.7 mm; used gas: nitrogen. The manufacture started with the preparation of the feed dispersion, which is the aqueous dispersion containing the active molecule and the excipient forming the micro delivery systems. To prepare the feed, an MI water solution (40 mL) containing an active amount ranging from 2 to 12 g was mixed with 40 mL of a water dispersion (30% *w*/*v*) of Eudraguard Biotic (12 g) under magnetic stirring. The feed was atomized and dried by the spray-dryer according to the conditions reported above.

### 4.4. Characterization of the Micro Delivery Systems

The drug loading (DL%) was calculated according to the below formula, and the amount of MI actually embodied in the microparticles was assessed by spectrophotometric analysis, according to the Megazyme K-INOSL kit.
$$\%DL=\frac{Mass\;of\;active\;in\;microparticles}{mass\;of\;microparticles}\times 100$$

The incorporation efficiency (%IE) was determined by accounting for the embodied myo-inositol within its respective micro delivery systems and the amount introduced in the feed of the spray-dryer.
$$\%IE=\frac{Mass\;of\;active\;in\;microparticles}{mass\;of\;active\;in\;the\;feed}\times 100$$

All experiments were carried out in triplicate. 

The Megazyme K-INOSL kit principle is described below: Myo-Inositol is oxidized by nicotinamide-adenine dinucleotide (NAD+) in the presence of myo-inositol dehydrogenase (lDH), leading to the formation of 2,4,6/3,5-pentahydroxycyclohexanone, reduced nicotinamide-adenine dinucleotide (NADH), and H+. A second reaction catalyzed by diaphorase is required, in which NADH reduces iodonitrotetrazolium chloride (INT) to an INT-formazan product, leading to a rapid and quantitative conversion of myo-inositol.

The amount of INT-formazan formed in this reaction is stoichiometric with the amount of myo-inositol. It is the INT-formazan which is measured by the increase in absorbance at 492 nm.

Particle size and morphology were measured by scanning electron microscopy (SEM) using a Phenom ProXSEM. All SEM analyses were performed at 25.0 °C ± 0.1 °C. The average diameter (d) ± standard deviation (SD) (mean ± SD) of the microparticles was determined from the mean value of 100 measurements using ImageJ (Madison, WI, USA, version 1.46 v). 

### 4.5. In Vitro Drug Release

The in vitro release of the MI embodied in the micro delivery systems was assessed using a dialysis bag method [30]. Microparticles equivalent to 100 mg were dispersed in 5 mL of 0.1 M HCl and poured into a cellulose ester dialysis membrane (cut off: 12–14 kDa) and sealed with appropriate universal closures (Spectrum chemical MFG Corp). This constitutes the donor compartment, which is then immersed in 95 mL of 0.1 M HCl (receiving compartment). The system (receiving + donating compartments) is thus maintained at pH = 1 under stirring and a constant temperature (100 rpm, 37 °C) for 2 h, making withdrawals at 10, 20, 30, 60, and 120 min. After 2 h, the pH of the receiving compartment is increased to pH 6.8 to mimic the first tract of the intestine. This pH is maintained for the following 4 h. It follows a further pH increase to pH 7.2 to mimic the environment of the colon until the last sampling is made, 24 h from the beginning of the experiment. A “myo-inositol assay kit K-INOSL” was used to determine the amount of released MI. Each sample was analyzed individually: in the first phase, the sample was incubated in a neutral environment in the presence of ATP and Hexokinase; then, it was brought to the basic environment, and other enzyme components of the kit were added and the absorbance at 492 nm was read using a PC Shimadzu Recording Spectrophotometer UV. Finally, the colorimetric reaction is triggered and, after an incubation period, the absorbance at 492 nm is read again: the reaction between the colored compound and MI is stoichiometric, enabling a calculation of the amount of MI present in the samples.

### 4.6. Ex Vivo Permeation Studies

Permeation studies were performed with Franz cells and porcine colon mucosae within an orbiting incubator at 37 °C (±1°). Sampling from the acceptor compartment was executed every 60 min for 10 h, plus a final sampling at 24 h. Each volume withdrawn for sampling was replaced by an equal amount of fresh buffer solution. Each sample was analyzed using the K-INOSL kit to determine the quantity of MI permeated in the recipient compartment (P) and the residual quantity in the donor compartment (R). The studies were carried out in triplicate and the experimental data are reported in concentration vs. time graphs, generating a permeation curve. The maximum concentration value measured (MI permeated in the receiving compartment + absorbed in the membrane) was equivalent to about 100% of loaded MI.

### 4.7. In Vivo Bioavailability and Ovarian Follicle Accumulation Studies

To assess the influence of the novel micro delivery system on the bioavailability of the myo-inositol (MI), in vivo experiments were carried out in two separate steps: evaluation of MI plasma concentrations at programmed time intervals over 24 h (bioavailability) after a single administration of the formulation (step 1), and evaluation of the MI ovarian follicle accumulation following daily administration for 1 week (step 2). 

Step 1: Fifteen female Wistar rats (250 ± 20 g) were randomly assigned to 3 groups (5 animals each): group 1: saline solution (sham group); group 2: myo-inositol (20 mg/kg); and group 3: microparticles containing myo-inositol (equivalent to 20 mg/kg of MI). One ml of saline solution or compound was administered by oral gavage and blood was collected via the tail vein under isoflurane anesthesia before (time 0) and at six-time points after administration (0.5, 1, 2, 6, 8, and 24 h). MI was dissolved in saline solution; microparticles containing myo-inositol were dispersed in food-grade mineral oil.

After blood collection, the animals were returned to their cages, where they remained with ad libitum access to food and water until the new time points of evaluation. After 24 h, the animals were euthanized under isoflurane anesthesia by cervical dislocation. Following the sacrifice, the ovaries were excised, weighed, and measured for biodistribution determination through a specific test for myo-inositol, the K-INOSL kit (Megazyme, Wicklow, Ireland). 

Step 2: Fifteen female Wistar rats (250 ± 20 g) were randomly assigned to 3 groups (5 animals each): group 4: saline solution (sham group); group 5: chronic myo-inositol (20 mg/kg); and group 6: chronic microparticles containing myo-inositol (equivalent to 20 mg/kg of MI). One ml of saline solution or compound was administered daily by oral gavage for 7 days. After 1 week, the animals were euthanized under isoflurane anesthesia by cervical dislocation. Following the sacrifice, the ovaries were excised, weighed, and measured for biodistribution determination through a specific test for myo-inositol: the K-INOSL kit (Megazyme, Wicklow, Ireland).

### 4.8. Evaluation of Myo-Inositol in Blood and Tissues

Plasma samples (approximately 60 µL) were obtained by centrifugation of the blood samples (100 µL) at 10,000 rpm for 5 min and immediately cryopreserved at −80 °C. To verify whether the presence of plasma proteins could interfere with the myo-inositol test, preliminary tests were carried out on blood samples from the control group. In this regard, plasma samples (25 µL) of the control group were deproteinized by adding 1N trifluoroacetic acid (100 µL), cooling in an ice bath for 10 min, and centrifugation at 4500 rpm for 10 min, and the supernatants were analyzed with the K-INOSL Megazyme kit after neutralization with a 1 N NaOH solution (100 µL). From the analysis of the results, no difference was observed with non-deproteinized blood samples. Consequently, the concentration of MI in each plasma sample was determined by directly analyzing 25 µL of plasma with the K-INOSL Megazyme kit. The ovarian follicles taken after 24 h (single administration) and 7 days (once-daily administration for 7 days) were homogenized with 800 µL of distilled water using an Ultraturrax homogenizer at 20,000 rpm for 5 min. The homogenates obtained were added to 1N trifluoroacetic acid (100 µL), placed in an ice bath for 10 min, and centrifuged at 4500 rpm for 10 min at 5 °C. The decanted supernatant (900 µL) was neutralized with a 1 N NaOH solution (100 µL) and finally filtered with 0.2-micron regenerated cellulose filters before analysis with the K-INOSL Megazyme kit, in 100 µL aliquots.

### 4.9. Statistical Analysis

The results obtained from multiple samples are expressed as mean ± standard deviation (SD). Statistical differences were analyzed by Student’s *t*-Test. A *p*-value < 0.05 was defined as the level of statistical significance.

## 5. Conclusions

In this work, we investigated a novel MI micro delivery system for the treatment of PCOS. Our results demonstrated that the micro delivery system herein presented can be easily prepared by spray-drying under mild conditions, featuring a controlled particle size below 20 µm and an optimal dosage form for oral administration, by using Eudraguard Biotic, an anionic copolymer of methyl acrylate, methyl methacrylate, and methacrylic acid, as a co-formulant in the spray-drying process. Thanks to the properties of this commercially available gastro-resistant excipient, the produced microparticles feature a delayed release and a preferential target release pattern of MI, principally in the intestine, thus controlling the bioavailability of the active molecule. The in vivo assessment of the MI-loaded micro delivery system demonstrated that the bioavailability and the follicular biodistribution increased by 50% and over 65%, respectively, when compared to the administration of the same dosage of pure MI. These features are extremely useful for the reduction in the therapeutic dosage of MI-based therapies, the eventual over-exposure of which has been shown to lead to gastric discomfort [24] upon chronic therapy. The results show that MI embodied in microparticles is absorbed more slowly and in a prolonged manner than pure MI, which, when administered at the same conditions, permeates the tissues in a maximum of 8–10 h, leading to a consequent faster elimination. The report of the respective AUC also predicts an increase in the overall bioavailability and half-life of myo-inositol due to the aforementioned prolonged absorption. From the evidence reported herein, it is possible to estimate a potential bioavailability increase of three times when the AUC of the myo-inositol is conveyed by our micro delivery system if compared to pure active MI.

## Figures and Tables

**Figure 1 ijms-25-03852-f001:**
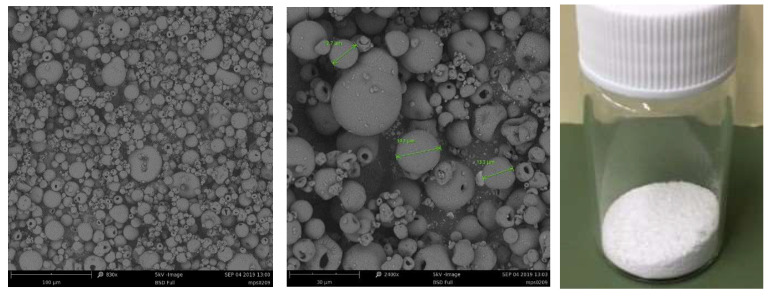
SEM image of the myo-inositol-loaded microparticles. Magnifications: 830× and 2400×, respectively. Size bars are 100 μm and 30 μm, respectively.

**Figure 2 ijms-25-03852-f002:**
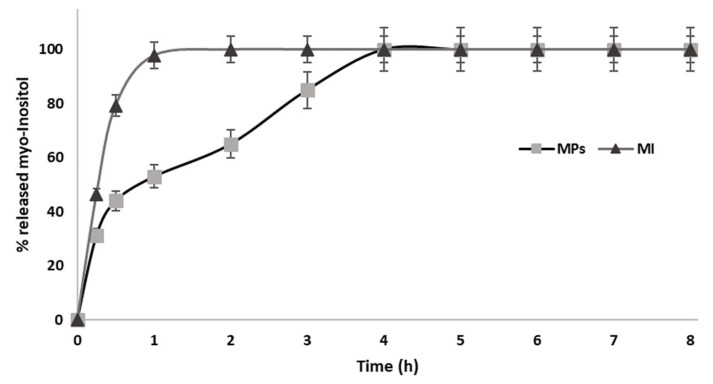
Myo-inositol released from the spray-dried batch n. 3 (MPs) and pure myo-inositol (MI). During the first 2 h, the pH of the receiving compartment was pH 1; then, it increased to pH 6.8 for the next 4 h. A further pH increase to pH 7.2 followed at 6 h.

**Figure 3 ijms-25-03852-f003:**
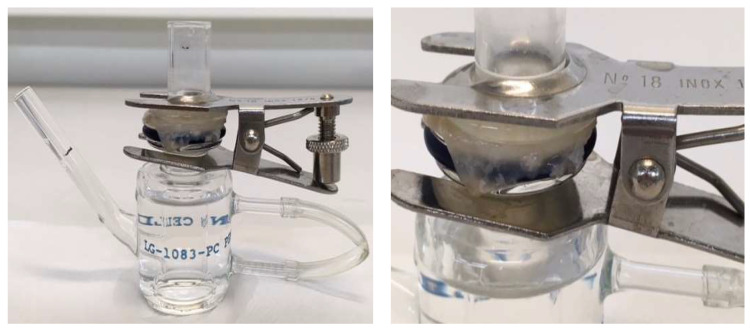
Franz cell set-up with the porcine colon mucosa, used for the permeation study.

**Figure 4 ijms-25-03852-f004:**
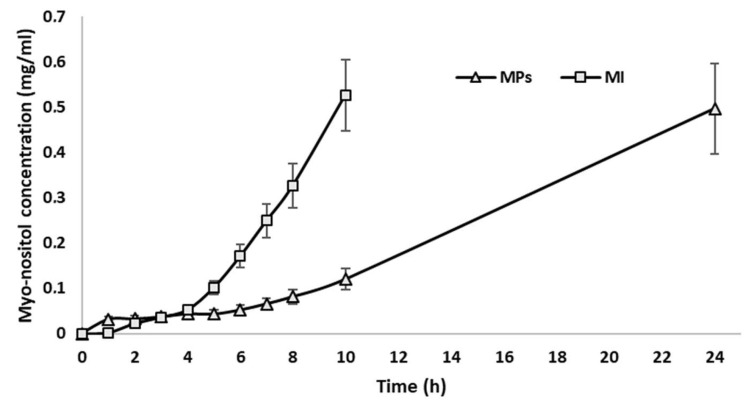
Permeation profile of MI through porcine colon mucosa; 0.5 mg/mL corresponds to 100% of loaded MI permeated in the receiving compartment. ∆ = MI-loaded microparticles (MPs), □ = pure myo-inositol (MI).

**Figure 5 ijms-25-03852-f005:**
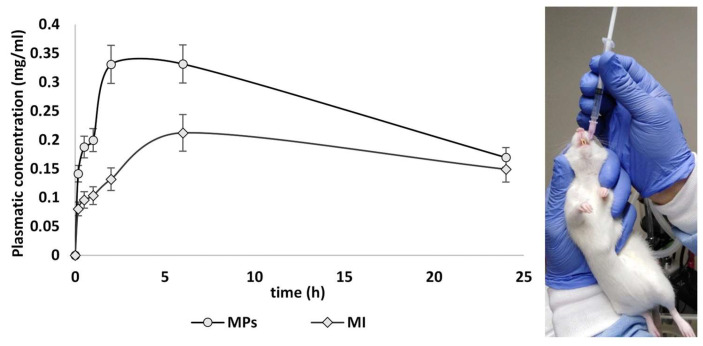
Graphical representation of bioavailability (plasma concentration vs. time) of MI in the different groups (2 and 3) after oral administration. Data are expressed as means ± SD of n = 5 for each group; *p* ˂ 0.05 versus MI group. Control curve was not reported because no MI was detected.

**Figure 6 ijms-25-03852-f006:**
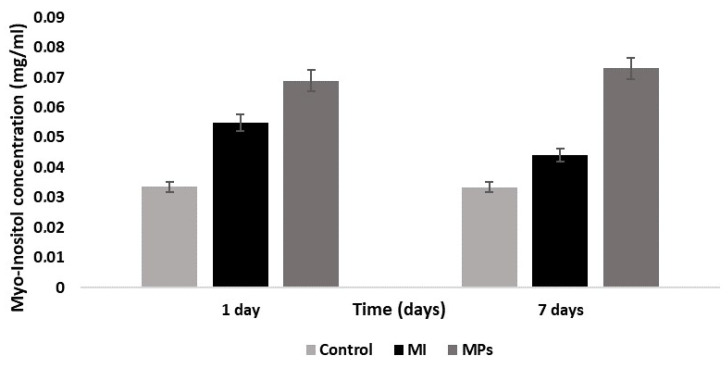
Histograms showing concentration of MI in the ovaries assessed after single administration of saline solution (control), MI, or MPs after single (1 day) and daily administration for 7 days. Data are expressed as means ± SD of n = 5 for each group; *p* ˂ 0.05 for each group.

**Table 1 ijms-25-03852-t001:** Summary of micro delivery system batches loaded with Myo-inositol (MI): drug loading (DL); incorporation efficiency (IE).

Batch No.	Eudragard/MI Weight Ratio	Yield%	DL%	EI%	% MI Released at pH 1 (2 h)
1	1	50 ± 3	47 ± 4	94 ± 1	90 ± 1
2	2	40 ± 2	25 ± 4	75 ± 2	80 ± 3
3	6	40 ± 2	14 ± 2	100 ± 2	65 ± 2

**Table 2 ijms-25-03852-t002:** AUC relates to the administration of pure MI or MI-loaded microparticles (MPs) and the bioavailability ratio given by the formula AUC MPs/AUC MI. All values are expressed in mg/mL·h. Maximum observed plasma concentration (C_max_; mg/mL) during the 0–24 h dosing interval; time to peak concentration (T_max_).

	MI	MPs	Ratios
AUC (mg/mL·h)	4.13 ± 0.24	6.24 ± 0.17	1.51
C_max_ (mg/mL)	0.21 ± 0.08	0.34 ± 0.08	-
T_max_ (h)	6 h	4 h	-

**Table 3 ijms-25-03852-t003:** Concentration values (average) of MI found in the tissues (ovarian follicles), expressed in mg/mL, upon the administration of pure MI or MI-loaded microparticles (MPs).

Sampling	Basal	MI	MPs	Ratio
1 day	0.033 ± 0.004	0.055 ± 0.002	0.069 ± 0.004	1.25
7 days	0.033 ± 0.003	0.044 ± 0.003	0.073 ± 0.002	1.66

## Data Availability

The data presented in this study are available on request from the corresponding author.

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
