# Peer review of "Anionic Methacrylate Copolymer Microparticles for the Delivery of Myo-Inositol Produced by Spray-Drying: In Vitro and In Vivo Bioavailability"

_ijms, 2024, doi:10.3390/ijms25073852_

Round 1
Reviewer 1 Report
Comments and Suggestions for Authors
The manuscript reports a new micro delivery system based on anionic methacrylate copolymer, able to improve the biological response of myo-inositol by daily oral administration was manufactured by spray drying. There are some questions want to discuss with authors.
1. Why the authors use pure Myo-inositol as the Comparator product? In the clinical aspects, the pure Myo-inositol can not be used. Usually, we would use a kind of medicine preparations of Myo-inositol, such as Trivitamins and Inositol Tablets, Inositol Nicotinate Tablets, Tabellae Inositoli Nicotinatis et al.
2. The Table 1. Showed the Summary of micro delivery system batches loaded with MI. % MI released at pH 1 (2h) are 90, 80, 65, which mean most of Myo-inositol were released and absorption, or maybe destructed in stomach. I wonder which part of the body is the target part of the Myo-inositol? In the line of 275, “4.5 In vitro drug release”, the authors employed a very complex method. Please explain why they did it? What are the references?
3. Most of the data are not presented as mean±SD.
4. In 4.7 In vivo bioavailability and ovarian follicles accumulation studies, there are only one dose of Myo-inositol. The experimental design needs to be more scientific about this problem. More often than not, there are at least two doses.
Comments on the Quality of English LanguageModerate editing of English language required.
Author Response
Reviewer 1
The manuscript reports a new micro delivery system based on anionic methacrylate copolymer, able to improve the biological response of myo-inositol by daily oral administration was manufactured by spray drying. There are some questions want to discuss with authors.
- Why the authors use pure Myo-inositol as the Comparator product? In the clinical aspects, the pure Myo-inositol can not be used. Usually, we would use a kind of medicine preparations of Myo-inositol, such as Trivitamins and Inositol Tablets, Inositol Nicotinate Tablets, Tabellae Inositoli Nicotinatis et al.
In this study, the authors want to highlight the advantages and benefits of a novel formulation produced by the spray-drying technique. This formulation is designed to provide a modified release profile of the drug and enhance the biological response to myo-inositol. Given the unique properties of the novel formulation herein discussed, it is important to establish its efficacy and superiority over the basic form of myo-inositol. By utilizing pure myo-inositol as the comparator, we could effectively showcase the enhanced performance of our formulation. We acknowledge that in clinical practice, myo-inositol is often administered in various pharmaceutical preparations, but it is also true that in the medicines mentioned above the release of myoinositol is neither controlled nor protected at the gastric level. However, for our study, we aimed to isolate the effects of our novel formulation against the baseline standard of myo-inositol.
- The Table 1. Showed the Summary of micro delivery system batches loaded with MI. % MI released at pH 1 (2h) are 90, 80, 65, which mean most of Myo-inositol were released and absorption, or maybe destructed in stomach. I wonder which part of the body is the target part of the Myo-inositol? In the line of 275, “4.5 In vitro drug release”, the authors employed a very complex method. Please explain why they did it? What are the references?
In Table 1, the authors summarized the three batches of microparticles that they produced based on different ratios between the drug and excipient. Based on the yield data of the production process, drug loading of the drug within the system, and the amount of drug released at acid pH, batch number 3 was selected because it was the one that ensured the least loss of drug at acid pH therefore at stomach level. It is known in biopharmaceutics that the target of the active ingredient is primarily the blood circulation. The more myo-inisitol reaches the bloodstream in the intact form, the higher the bioavailability and consequently the probability that the active ingredient reaches the target site, in this case the ovarian follicles. Since the aim of this work is the creation of a new drug delivery system that presents a modified/delayed release profile, able to improve bioavailability of Myo-inositol.
Regarding the type of method used for the drug release study, the authors added a citation (31) in the line cited by the reviewer, to better justify the experimental choice. The use of dialysis membranes to mimic a donor compartment within which the formulation is present and which therefore allows the diffusion of the drug towards a recipient compartment is a method commonly used to mimic the diffusion processes that occur in physiological compartments in the human body.
- Most of the data are not presented as mean±SD.
The authors thank the reviewer for the comment and as request SD values were added where missing.
- In 4.7 In vivo bioavailability and ovarian follicles accumulation studies, there are only one dose of Myo-inositol. The experimental design needs to be more scientific about this problem. More often than not, there are at least two doses.
The authors are aware of the fact that the use of different doses would certainly have increased the significance of the bioavailability results, but it is also true that the final result would not have changed. On the other hand, testing a second dose would also have meant the use of double the number of animals which, for ethical and regulatory reasons, the authors preferred not to use. It is the authors opinion that a single dose of administered drug was sufficient to demonstrate that the new release system designed and developed by spray-drying technique is more efficient, as it allows a considerable increase in bioavailability, if compared to the same administered dose of pure drug.
Reviewer 2 Report
Comments and Suggestions for Authors
The article entitled “Anionic methacrylate copolymer microparticles for the delivery of myo-inositol produced by spray-drying: in vitro and in vivo bioavailability” is a research work specifically focused on a new micro delivery system based on anionic methacrylate copolymer, able to improve the biological response of myo-inositol by daily oral administration was manufactured by spray drying. Authors are encouraged to consider the following comments and suggestions for further refinement of their work
1. Authors should add the mathematic data in the abstract such as Particle size, zeta potential, % release, and % bioavailability for a clear understanding of the article's outcome.
2. Show the Myo-inositol (MI) activity model as an image pathway form for a clear understanding
3. What is the reason for the selection of an anionic carrier for the spray drying formulation?
4. Authors should provide an appropriate reference for the Claim that “traditional MI supplement formulations are unable to ensure reproducibility in terms of oral absorption and bioavailability”. More what is the Bioavailability of the traditional MI supplement? What are different Tradition formulations available for the MI supplement? What is the high dose which produces the gastrointestinal side effects? The claims of the failure of the traditional MI supplement formulations by Authors need to be supported by references as well as more scientific data and examples.
5. The actual aim and objective of the research work is not clear in the research introduction. The author should provide a clear novelty, main, and objective statement in the introduction section
6. Spray drying is high temperature and air-flow-dependent parameters. Here authors use Inlet T: 110°C; Outlet T: 60°C; 100% suction; Feed pump, 15%; Atomizer nozzle: 0.7 mm; Used gas: Nitrogen. On which basis do authors optimize the process parameters for the study?
7. Provide a detailed method of the estimation of the drug loading and incorporation efficiency (%IE) along the spectrophotometric analysis parameters
8. The stomach pH is around 1.5 to 3.5, then Why authors select pH 1 for the first 2 hr Release study, provide an explanation with a reference
9. “ MI water solution 253 (40 mL) containing an active amount ranging from 2 to 12 gr, was mixed with a water dispersion of Eudraguard Biotic (40 mL, 30% w/v as provided by the manufacturer) under magnetic stirring”. Why authors select to check the different MI concentrations rather than the Eudragard in this study? Or maybe the authors need to improve the language to make it clear.
10. As per Table 1 the % DL reduces with the higher weight ratio, The Authors need to explain the reason for the Results outcome. Moreover, what is the %IE of the different batches?
11. Authors use 3 different release conditions pH=1, pH 6.8 to mimic the first tract of the intestine. And pH 7.2 to mime the environment of the colon. But In the case of the Release Profile authors only showed the Release in pH 1. Provider the full release profile with the proper explanation of the in Vitro release data
12. As per Table 1, The DL for each sample is different, What is the equivalent weight authors use for the in vitro study for each formulation?
13. “The release studies revealed the relationship between the above-mentioned ratio and release performance and made us select batch n. 3”. If authors selected batch No. 3 based on the control release why the yield and the DL% not being considered for the batch selection? The author's justification for the selection of the batch is unrealistic and non-scientific.
14. Authors used an Atomizer nozzle: of 0.7 mm but an average size of 13.5 ± 5.7 microns, Explain the reason. Moreover, the SEM images (30 microns) indicated larger particle sizes with no size uniformity or homogeneous condition. SEM image of e 100 microns indicates that the SD of the Particle size is larger than 5.7 microns. Authors should Correlate they result statement with image 1.
15. Authors claim they assessed in vitro by a pH jump test. In the whole article, they did not provide the Release of the different pH
16. The Release profile in Figure 2 of spray-dried batch n. 3 (MPs) and pure myo-inositol (MI) have the same release at 8 hr. How do the authors claim the sustained release of the spray dry formulation?
17. Provide the apical-to-basal Papp Value for the permeation model rather than the AUC data to clear permeability enhancement.
18. The animal data in Table 2 lack statistical significance. Provide the unit for the AUC
19. Provide the histopathological images of the ovarian follicles
20. Provide a clear conclusion for the research outcome
21. The English language needed a professional revision
Comments on the Quality of English LanguageExtensive editing of the English language required
Author Response
Reviewer 2
The article entitled “Anionic methacrylate copolymer microparticles for the delivery of myo-inositol produced by spray-drying: in vitro and in vivo bioavailability” is a research work specifically focused on a new micro delivery system based on anionic methacrylate copolymer, able to improve the biological response of myo-inositol by daily oral administration was manufactured by spray drying. Authors are encouraged to consider the following comments and suggestions for further refinement of their work
- Authors should add the mathematic data in the abstract such as Particle size, zeta potential, % release, and % bioavailability for a clear understanding of the article's outcome.
According to the reviewer's comment, the authors added more mathematical data in the abstract (red text).
- Show the Myo-inositol (MI) activity model as an image pathway form for a clear understanding
The authors would like to please the reviewer by adding a descriptive figure of the mechanism of action of myo-inositol. However, the aforementioned mechanism is already well described in the references included in the introduction of the manuscript, which already contain countless descriptive schemes. Therefore, the authors considered that reproducing a new image would not be useful in this case, since the present study is independent of the mechanism of action of the active ingredient.
- What is the reason for the selection of an anionic carrier for the spray drying formulation?
As the authors wrote in the introduction section, Eudragard Biotic is an anionic copolymer of methyl acrylate, methyl methacrylate, and methacrylic acid, in a 7:3:1 ratio respectively, approved by the European Commission as a safe food additive and is employed as a coating material for colon/delayed-release in gastro-resistant oral dosage forms. Starting from the assumption that the authors wanted to use an excipient recognized as a GRAS (safe excipient), the choice to use this excipient was also supported by its chemical characteristics which express a pH-dependent solubility. In this work these properties have been exploited to create a modified release system. The references that confirm these properties are 26, 27 and 30.
- Authors should provide an appropriate reference for the Claim that “traditional MI supplement formulations are unable to ensure reproducibility in terms of oral absorption and bioavailability”. More what is the Bioavailability of the traditional MI supplement? What are different Tradition formulations available for the MI supplement? What is the high dose which produces the gastrointestinal side effects? The claims of the failure of the traditional MI supplement formulations by Authors need to be supported by references as well as more scientific data and examples.
The authors agree with reviewer that a more reference support is needed at this regard. Therefore, new references have been added in the introduction, as below listed and reported in the revised text as references 21-23:
Dinicola, S.; Minini, M.; Unfer, V.; Verna, R.; Cucina, A.; Bizzarri, M. Nutritional and Acquired Deficiencies in Inositol Bioavailability. Correlations with Metabolic Disorders. Int. J. Mol. Sci. 2017, 18, 2187.
Gianfranco Carlomagno , Sara De Grazia , Vittorio Unfer & Fedele Manna (2012) Myo-inositol in a new pharmaceutical form: a step forward to a broader clinical use, Expert Opinion on Drug Delivery, 9:3, 267-271.
Simone Garzon, Antonio Simone Laganà & Giovanni Monastra (2019) Risk of reduced intestinal absorption of myo-inositol caused by D-chiro-inositol or by glucose transporter inhibitors, Expert Opinion on Drug Metabolism & Toxicology, 15:9, 697-703.
- The actual aim and objective of the research work is not clear in the research introduction. The author should provide a clear novelty, main, and objective statement in the introduction section
According to the reviewer's comment, the authors explain better the aim and the novelty of the work in the introduction section (red text, lines 84-97 of revised manuscript).
- Spray drying is high temperature and air-flow-dependent parameters. Here authors use Inlet T: 110°C; Outlet T: 60°C; 100% suction; Feed pump, 15%; Atomizer nozzle: 0.7 mm; Used gas: Nitrogen. On which basis do authors optimize the process parameters for the study?
The choice of the operating parameters of the spray dryer was made on the authors' great knowledge of the widely used production technique and based on citations numbers 32, 33, 34, and 35 present in the manuscript. The optimization of process parameters is centred on varying the Eudragard/MI weight ratio and if necessary the spray dryer operating parameters, such as temperature, air- and feed-flow values. Subsequently various production attempts are carried out, until the product with the best qualities in terms of yields, DL and release profile is obtained. For better clarify this aspect to the readers, the above sentences was added in the Results section.
- Provide a detailed method of the estimation of the drug loading and incorporation efficiency (%IE) along the spectrophotometric analysis parameters
The authors described the experimental procedure for calculating drug loading and incorporation efficiency in the materials and methods section, however for the determination of inositol an experimental kit from the Megazyme company was used and the protocol indicated by the kit was followed (K-INOSL kit). About this point more detail were added in the experimental section, paragraph 4.4.
- The stomach pH is around 1.5 to 3.5, then Why authors select pH 1 for the first 2 hr Release study, provide an explanation with a reference
The pH of solutions mimicking gastric fluids is established by a convention and reported as pH 1 in all official pharmacopoeias. For this reason, the authors decided to use pH 1 for the first two hours of the experiment to mimic the gastric environment. However, the authors added the right reference for this experiment (30).
- “ MI water solution 253 (40 mL) containing an active amount ranging from 2 to 12 gr, was mixed with a water dispersion of Eudraguard Biotic (40 mL, 30% w/v as provided by the manufacturer) under magnetic stirring”. Why authors select to check the different MI concentrations rather than the Eudragard in this study? Or maybe the authors need to improve the language to make it clear.
According to reviewer suggestion, the above sentence was rewritten as follow in the revised manuscript: “with 40 ml of a water dispersion 30% w/v of Eudraguard Biotic (12 gr)”.
- As per Table 1 the % DL reduces with the higher weight ratio, The Authors need to explain the reason for the Results outcome. Moreover, what is the %IE of the different batches?
As the authors wrote in the manuscript, the drug loading of MI ranged between 14 and 47% because in the different batches, the Eudragard/MI weight ratio varied from 6 to 1 respectively, so when the amount of MI decreased also the drug loading decreased. Furthermore, the authors added the EI% values in Table 1 to better understand how much of the drug used in the formulation process is actually loaded inside the microparticles, according with the formulas used for the calculations and reported in the materials and methods section.
- Authors use 3 different release conditions pH=1, pH 6.8 to mimic the first tract of the intestine. And pH 7.2 to mime the environment of the colon. But In the case of the Release Profile authors only showed the Release in pH 1. Provider the full release profile with the proper explanation of the in Vitro release data
As the authors wrote in the Materials and methods section: “ In vitro release of the MI embodied in the micro delivery systems was assessed using a dialysis bag method [28]. Microparticles equivalent to 100 mg were dispersed in 5 ml of 0.1M HCl and poured into a cellulose ester dialysis membrane, cut-off 12–14 kDa, sealed with appropriate universal closures (Spectrum chemical MFG Corp). This constitutes the donor compartment, which is then immersed in 95 ml of 0.1M HCl (receiving compart-ment). The system (receiving + donating compartments) is thus maintained at pH=1 under stirring and constant temperature (100 rpm, 37°C°) for 2h, making withdrawals at 10, 20, 30, 60, and 120 minutes. After 2h, the pH of the receiving compartment is increased to pH 6.8 to mimic the first tract of the intestine. This pH is maintained for the following 4h. It follows a further pH increase to pH 7.2 to mime the environment of the colon until the last sampling is made, 24h from the beginning of the experiment.”. In order to make clearer this point the legend of Figure 2, showing the full release profile of the in vitro study, was implemented with pH change details.
- As per Table 1, The DL for each sample is different, What is the equivalent weight authors use for the in vitro study for each formulation?
For the in vitro study, the authors compared the dissolution profile of pure active ingredient with the formulation batch number 3. To be able to normalize and compare the results obtained with each other, a quantity by weight of microparticles equal to that of the pure drug was used. Therefore, for 100 mg of microparticles having a drug loading of 14%, i.e. containing 14 mg of the active ingredient, they were compared with the release profile of 14 mg of pure myo-inositol.
- “The release studies revealed the relationship between the above-mentioned ratio and release performance and made us select batch n. 3”. If authors selected batch No. 3 based on the control release why the yield and the DL% not being considered for the batch selection? The author's justification for the selection of the batch is unrealistic and non-scientific.
In agreement with the reviewer, the authors clarified and rewrote the concept better (paragraph “Results”, lines 108-114), highlighting how all the parameters together led to the selection of batch 3 for subsequent studies.
- Authors used an Atomizer nozzle: of 0.7 mm but an average size of 13.5 ± 5.7 microns, Explain the reason. Moreover, the SEM images (30 microns) indicated larger particle sizes with no size uniformity or homogeneous condition. SEM image of e 100 microns indicates that the SD of the Particle size is larger than 5.7 microns. Authors should Correlate they result statement with image 1.
Authors thank the reviewer for the comments. We appreciate the opportunity to clarify certain aspects of our study. Regarding the size of the Atomizer nozzle, it's important to note that in techniques such as spray-drying, the size of the atomizer nozzle doesn't precisely dictate the final size of the microparticles produced. Instead, it sets a maximum limit on particle size. Specifically, microparticles larger than 700 micrometers cannot be obtained, but smaller sizes are achievable. Typically, with this type of instrument, the particle size range falls between 5 and 50 μm. Our microparticles fall within this achievable size range. Additionally, the SEM images were provided with differing scale bars to offer both a panoramic view of the sample and a more zoomed-in perspective. This was done to provide a comprehensive understanding of the sample morphology. Furthermore, the analysis for determining the average particle diameter was conducted using ImageJ software on a sample of 100 microparticles. We have included this additional information in the corresponding Materials and Methods section of the manuscript.
- Authors claim they assessed in vitro by a pH jump test. In the whole article, they did not provide the Release of the different pH
The entire in vitro release study is shown in Figure 2 in a cumulative graph that shows the release profile of the active ingredient during the time and in different pH conditions. Referring to the materials and methods section: “ In vitro release of the MI embodied in the micro delivery systems was assessed using a dialysis bag method [28]. Microparticles equivalent to 100 mg were dispersed in 5 ml of 0.1M HCl and poured into a cellulose ester dialysis membrane, cut-off 12–14 kDa, sealed with appropriate universal closures (Spectrum chemical MFG Corp). This constitutes the donor compartment, which is then immersed in 95 ml of 0.1M HCl (receiving compartment). The system (receiving + donating compartments) is thus maintained at pH=1 under stirring and constant temperature (100 rpm, 37°C°) for 2h, making withdrawals at 10, 20, 30, 60, and 120 minutes. After 2h, the pH of the receiving compartment is increased to pH 6.8 to mimic the first tract of the intestine. This pH is maintained for the following 4h. It follows a further pH increase to pH 7.2 to mime the environment of the colon until the last sampling is made, 24h from the beginning of the experiment.”.
- The Release profile in Figure 2 of spray-dried batch n. 3 (MPs) and pure myo-inositol (MI) have the same release at 8 hr. How do the authors claim the sustained release of the spray dry formulation?
We understand the concern about the apparent similarity in release profiles between spray-dried batch n. 3 (MPs) and pure myo-inositol (MI) at 8 hours. However, it's crucial to consider the context in which we define the sustained release of the spray-dried formulation. The term "sustained release" or "modified release" was employed by comparing the release profile of the microparticles with that of the equivalent amount of pure active ingredient. In this case, pure myo-inositol dissolves completely within 1 hour under acidic pH conditions, failing to reach the intestinal target site effectively. On the other hand, the microparticle system exhibits minimal myo-inositol release during the initial two hours under acidic pH conditions. Only after the pH transition occurs, reaching the intestinal target site, does the microparticle system gradually release myo-inositol, ultimately achieving 100% release. Therefore, while the release profiles at 8 hours may appear similar, it's the overall sustained release behavior throughout the experiment and the targeted delivery achieved that characterize the spray-dried formulation as exhibiting sustained release properties.
- Provide the apical-to-basal Papp Value for the permeation model rather than the AUC data to clear permeability enhancement.
We appreciate your suggestion to provide the apical-to-basal Papp value for the permeation model rather than the AUC data. However, we would like to clarify that our ex-vivo permeation study was conducted using tissue samples obtained from animals, rather than utilizing a cell culture model in the laboratory. As such, we did not calculate any Papp values in our study. The decision to use ex-vivo tissue samples was made to better mimic the physiological conditions and complexity of the in vivo environment. While we acknowledge that Papp values can provide valuable insights into permeability enhancement, our study focused on assessing the overall permeation behavior of the formulation rather than quantifying specific permeability parameters.
- The animal data in Table 2 lack statistical significance. Provide the unit for the AUC
The authors thank the reviewer for bringing these oversights to our attention which has enabled us to improve the quality and completeness of our manuscript.
- Provide the histopathological images of the ovarian follicles
The authors want to clarify that the histopathological analysis of the ovarian follicles is not the subject of this study, as the main aim was to evaluate the changes in terms of bioavailability and localized accumulation of the drug of the new micro delivery system produced, therefore it was not any histopathological analysis performed.
- Provide a clear conclusion for the research outcome
The authors clarified better the research outcomes in all the sections of the manuscript.
- The English language needed a professional revision
According to the reviewer's comment, the authors revised the English language of the manuscript.
Round 2
Reviewer 2 Report
Comments and Suggestions for Authors
Revised
Comments on the Quality of English LanguageModerate